# Prognostic Role of Molecular and Imaging Biomarkers for Predicting Advanced Hepatocellular Carcinoma Treatment Efficacy

**DOI:** 10.3390/cancers14194647

**Published:** 2022-09-24

**Authors:** Lucia Cerrito, Maria Elena Ainora, Carolina Mosoni, Raffaele Borriello, Antonio Gasbarrini, Maria Assunta Zocco

**Affiliations:** 1Department of Internal Medicine and Gastroenterology, Fondazione Policlinico Universitario Agostino Gemelli IRCCS, Catholic University of Rome, 00168 Rome, Italy; 2CEMAD Digestive Disease Center, Fondazione Policlinico Universitario Agostino Gemelli IRCCS, Catholic University of Rome, 00168 Rome, Italy

**Keywords:** hepatocellular carcinoma, prognosis, biomarkers, alpha-fetoprotein, systemic treatment

## Abstract

**Simple Summary:**

Molecular biomarkers play a marginal role in clinical practice for hepatocellular carcinoma (HCC) diagnosis, surveillance and treatment monitoring. Radiological biomarker: alpha-fetoprotein is still a lone protagonist in this field. The potential role of molecular biomarkers in the assessment of prognosis and treatment results could reduce the health costs faced by standard radiology. The majority of efforts are oriented towards early HCC detection, but the field faces an important challenge to find adequate biomarkers for advanced HCC management.

**Abstract:**

Hepatocellular carcinoma (HCC) is the sixth most common malignancy worldwide and the fourth cause of tumor-related death. Imaging biomarkers are based on computed tomography, magnetic resonance, and contrast-enhanced ultrasound, and are widely applied in HCC diagnosis and treatment monitoring. Unfortunately, in the field of molecular biomarkers, alpha-fetoprotein (AFP) is still the only recognized tool for HCC surveillance in both diagnostic and follow-up purposes. Other molecular biomarkers have little roles in clinical practice regarding HCC, mainly for the detection of early-stage HCC, monitoring the response to treatments and analyzing tumor prognosis. In the last decades no important improvements have been achieved in this field and imaging biomarkers maintain the primacy in HCC diagnosis and follow-up. Despite the still inconsistent role of molecular biomarkers in surveillance and early HCC detection, they could play an outstanding role in prognosis estimation and treatment monitoring with a potential reduction in health costs faced by standard radiology. An important challenge resides in identifying sufficiently sensitive and specific biomarkers for advanced HCC for prognostic evaluation and detection of tumor progression, overcoming imaging biomarker sensitivity. The aim of this review is to analyze the current molecular and imaging biomarkers in advanced HCC.

## 1. Introduction

Hepatocellular carcinoma (HCC) is the fifth most common malignancy worldwide and, in terms of mortality, represents the fourth cause of tumor-related death (second in males and the sixth in females) [1]. In the United States, HCC has been the cause of cancer-related death with the fastest rise in the last few years [2]. The annual incidence of HCC is constantly growing and, by 2025, more than 1 million new annual cases are expected [3].

It represents the main primary malignancy of the liver, followed by intrahepatic cholangiocarcinoma (CCC) [4].

The 5-year HCC survival rate is about 18%, appointing it as the second most lethal tumor after pancreatic cancer [5]. Advanced-stage HCC, which is usually defined by the Barcelona clinic liver cancer (BCLC) C-D stages and is characterized by a performance status according to the Eastern Cooperative Oncology Group (ECOG) of 1–2 (or 3–4 in BCLC stage D), vascular invasion or extrahepatic spread [5]. Advanced HCC accounts for more than 35% of newly diagnosed HCC cases [6], with a 5-year survival rate of 2.5% [7], a median survival rate of 11 months in cases of extrahepatic spread, and 7 months with neoplastic main portal vein thrombosis [2].

Systemic therapy is the only therapeutic approach for advanced HCC, with atezolizumab-bevacizumab as the current first-line therapeutic choice, along with Lenvatinib or Sorafenib if the principal option is not feasible [8].

Despite the recent noteworthy improvements in the development of new therapies for advanced HCC, there is still a lack of biomarkers able to predict the treatments’ efficacy or identify subgroups of patientswhich would benefit from a certain drug. Alpha-fetoprotein (AFP) confirmation is the only molecular biomarker with recognized prognostic value [3].

This review focuses on the potential role of molecular and radiological biomarkers in predicting the therapeutic efficacy and survival in advanced HCC patients, with a role in identifying specific subgroups of patients who could benefit from definite medical treatments or predicting early therapeutic failure to optimize the standard of care in these patients.

## 2. Biomarkers in HCC

A biomarker is a biochemical or clinical factor that provides reliable clues about disease burden and outcome, especially in response to specific treatments with relevant prognostic and predictive implications. The histotype, the prevalence of a tumor and the existence of efficacious treatments are important elements affecting the potential value of a biomarker [9].

There are several parameters that can be considered valuable prognostic parameters for HCC: serum proteins such as AFP and its glycosylated form (Lens culinaris agglutinin-reactive AFP or AFP-L3), des-γ-carboxy prothrombin (DCP, or prothrombin induced by vitamin K absence II [PIVKA II]), tumor number and dimension, status of surgical margins, neoplastic stage, vascular invasion, portal hypertension, and Child-Pugh score [10].

The traditional and well-proven imaging biomarkers are widely recognized as the gold standard in HCC management (diagnosis and treatment monitoring), they are represented by ultrasound (US), contrast-enhanced ultrasound (CEUS), computed-tomography (CT), and magnetic resonance imaging (MRI). The last two methods in particular are the gold standard for the assessment of neoplastic burden after treatment based on the standardized rules applied to imaging techniques: the Response evaluation criteria in solid tumors (RECIST) and their modified versions (modified-RECIST and RECIST 1.1) [11].

Even adverse events represent potential response biomarkers during systemic treatment (especially tyrosine-kinase inhibitors). It has been reported that hypertension, diarrhea and hand-foot skin reaction are correlated to sorafenib clinical efficacy [12].

Due to the presence of chronic liver disease at different stages (especially cirrhosis) in patients developing HCC, the relevant clinical biomarkers used as prognostic factors at the basis of the Child-Pugh score are liver function (total bilirubin, albumin, International Normalized Ratio—INR) and portal hypertension with its possible complications (ascites, encephalopathy) [13].

Regarding molecular biomarkers, it is important to underline that the wide heterogeneity of HCC’s molecular features has potentially relevant therapeutic and prognostic implications. In fact, HCC represents a complex environment involving both neoplastic and non-tumor cells (mainly belonging to the immune system). The reciprocal interplay between these elements is crucial to enhance the success of the different therapeutic options available.

Despite the advances in terms of therapeutic options, serum AFP is still the main molecular biomarker used for diagnostic and prognostic purposes, and treatment response monitoring. Nevertheless, its role is still debated, especially in HCC surveillance and diagnosis [9], due to its reliance on HCC prevalence in the studied population [14], suboptimal sensitivity (53% with cut-off 20 ng/mL), and limited specificity (AFP positive in 60–80% HCCs). Moreover, the presence of false-positives reduces the possibility of detecting early-HCC in both benign and malignant gastrointestinal disorders (e.g., acute and chronic hepatitis, intrahepatic cholangiocarcinoma) [15,16]. Another tough point is the variability of AFP specificity and sensitivity relatively to the chosen cut-off: the highest sensitivity (60%) in early HCC diagnosis is reached with AFP > 20 ng/mL (more than with higher cut-offs); specificity grows with higher cut-off values [9].

In the last decades no significant improvements have been achieved in the field of molecular biomarkers for HCC that still have weak roles in clinical practice. Among them the fucosylated fraction of AFP, proteoglycans such as glypican 3 and versican, DCP and PIVKA II have been principally investigated for early-stage HCC detection, treatment response monitoring and tumor prognosis evaluation [11,17,18].

Biochemical biomarkers have also been gradually included in some clinical staging algorithms; however, these still remain at the periphery of both the diagnostic and prognostic processes.

The routinely applied therapeutic/prognostic decisional algorithm Barcelona clinic liver cancer (BCLC), traditionally based on tumor burden, liver function (Child-Pugh, model of end-stage liver disease (MELD), and albumin-bilirubin “ALBI” score), and patient’s physical status (ECOG), in its most recent version introduced for the first time AFP as a criterium for liver transplant evaluation and to assess the eligibility to systemic treatment with ramucirumab [8]. Other renowned scores including AFP are the cancer of the liver Italian program (CLIP) (along with the Child-Pugh score, neoplastic mass morphology, and portal vein thrombosis) [19], Chinese University Prognostic Index (CUPI) (including TNM staging, ascites, symptoms, alkaline phosphatase, and bilirubin) [20], and the staging system “Groupe d’Etude et de Traitement du Carcinome Hépatocellulaire-GRETCH” (considering also the Karnofsky index, serum bilirubin, alkaline phosphatase, and ultrasonographic portal obstruction) [21].

Nowadays, the main unmet need in HCC management remains the discovery of sensitive and specific biomarkers providing precocious information about the progression and prognosis of advanced HCC compared to traditional radiological biomarkers.

The recently developed the BALAD score, based on the combination of biochemical values (albumin, serum bilirubin) and a panel of three biomarkers (AFP > 400 ng/mL, Lens culinaris agglutinin-reactive AFP [AFP-L3] > 15%, and DCP > 100 mAU/mL) have been proposed as a method to detect the worse prognosis, reflecting neoplastic progression and metastasis [22]. However, the initial validation studies did not show the superiority of this score compared to other systems without biomarkers. Moreover, the absence of support from traditional imaging features lead to possible treatment selection bias, thus these scores were disregarded in clinical practice [23].

## 3. Serum Biomarkers

The serum is the most accessible source of biomarkers and a simple and non-invasive method to monitoring HCC treatment. Serum biomarkers can be categorized as protein markers, growth factors and circulating nucleic acid (Figure 1).

### 3.1. Protein Markers

AFP is still the principal and most reliable serum biomarker for HCC diagnosis, treatment management, and prognosis evaluation despite numerous attempts to investigate other possible molecules [17]. The prognostic role of elevated AFP in advanced HCC has been assessed in several clinical trials concerning first- or second-line systemic therapy [24,25,26,27,28,29].

It has been demonstrated that an AFP reduction of more than 20% after 4 weeks of antiangiogenetic treatment is an important predictor of a patient’s response to therapy and survival [30,31].

Similar results were obtained in a retrospective study performed in patients with advanced HCC undergoing a combined therapy of sorafenib and transarterial chemoembolization (TACE). A higher median overall survival (OS) was found in the group of patients showing AFP reduction at an earlier time point (12.8 vs. 6.4 months; *p* = 0.001) compared to the results granted by the RECIST and mRECIST criteria [32].

In a multicentre phase III trial plasma levels of AFP demonstrated a prognostic role in identifying patients who could benefit from second-line treatment with ramucirumab (after sorafenib). An improvement in median overall survival (OS 8.5 vs. 7.3 months; hazard ratio, HR 0.71; *p* = 0.0199) and progression free survival (PFS) (2.8 months vs. 1.6 months; HR 0.452, *p* < 0.0001) was observed in patients with AFP ≥ 400 ng/mL treated with ramucirumab compared to the placebo [27,33].

Even in the encouraging field of immune checkpoint inhibitors (ICI) for the treatment of advanced HCC, AFP confirms its role as in predicting treatment response and OS. Shao et al. recently reported that an AFP reduction >20% in the first 4 weeks of therapy is associated to therapeutic efficacy [34].

Lee et al. presented the “10-10 rule” for the prediction of ICI responses based on a baseline AFP serum level of ≥10 ng/mL and a subsequent 10% decrease after 4 weeks of treatment. They evaluated 95 patients with advanced HCC and a baseline AFP of ≥10 ng/mL receiving anti-programmed cell death-1 (PD-1) antibodies (nivolumab or pembrolizumab), and identified a >10% AFP reduction after 4 weeks of treatment as an independent predictor of the best objective response (objective response rate, ORR 24.4%; odds ratio, OR: 7.259; *p* = 0.001) [35]. They also investigated the expression of PD-ligand 1 (PD-L1), observed in neoplastic cells and inflammatory cells belonging to the intratumoral environment and used a conventional response-to-treatment biomarker for other cancers [36]. Despite the small group of patients evaluated, the authors observed a greater disease progression in patients with a PD-L1 expression of <1%, suggesting a possible future use of this biomarker in the detection of HCC response to ICI [35].

Several attempts were made in the past to find other potential biomarkers to define responses to treatment in advanced HCC and the subsequent outcomes. This was driven by the need to find a tool with adequate specificity and sensitivity, suitable to establish a patient-tailored plan for treatment, and prognostic foresights in terms of response and survival, especially after the approval of ICI for HCC. The discovery of circulating biomarkers sufficiently predictive of PFS and OS could allow the selection of the best treatment option for advanced HCC.

Nakamura et al. examined the role of DCP, a non-functional prothrombin precursor, in 1377 patients with HCC. It showed better performance in the detection of intermediate/advanced HCC compared to small HCCs. The area under the receiver operating characteristic (AUROC) curve for DCP was significantly lower than that of AFP in small HCCs (<3 cm), but was significantly higher in large HCCs (>5 cm) [37]. Similar results were obtained in another study by Koike et al. performed on 227 HCC patients, where DCP serum levels demonstrated a strong prognostic role in the detection of portal vein invasion, followed by tumor histological grade and differentiation [38].

Soluble immune checkpoint proteins are involved in both stimulating and inhibiting the factors responsible for T-cell activation and proliferation in the defense against tumors. Dong et al. examined plasma levels of 16 soluble immune checkpoint proteins in 53 patients with advanced HCC treated with sorafenib. The best biomarker observed was the soluble B and T lymphocyte attenuator (sBTLA), involved in the host’s anti-tumoral immune response, that proved to be an independent predictor of poor OS by a multivariate analysis (OS was 2-times longer with high levels of sBTLA; *p* < 0.05) [39].

Among serum proteins, annexin A2, a protein involved in tumor progression (cellular proliferation, tissue aggressiveness and metastatic process) represents an interesting potential biomarker in advanced HCC [40]. Li et al. demonstrated that the association of annexin A2 with engulfment and cell motility protein 1 (ELMO1)/dedicator of cytokinesis (Dock180) regulates actin polymerization, chemotaxis, cell migration and the metastatic process in patients with advanced HCC. Annexin A2 expression in neoplastic tissues was directly proportional with lymphonodal involvement and metastasis [41].

Another important serum protein, survivin, a potent anti-apoptotic protein with a key role in cellular proliferation and stromal neoangiogenesis seems to be associated to tumoral aggressiveness in various cancers, including HCC. Its malignant biological behavior affects prognosis through a reduced sensitivity to chemotherapy and radiotherapy and, for these reasons, could be an interesting target for the development of new HCC therapies [42].

A small study by Oh et al. investigated the role of a-disintegrin-and-a-metalloprotease-9 (ADAM9) as a potential biomarker for HCC clinical responses to immunotherapy in patients with advanced HCC [43].

ADAM9 is involved in cellular adhesion and migration, shedding of membrane proteins, and proteolysis of the extracellular matrix [44]. In physiological situations, major histocompatibility protein (MHC) class I-related chain A (MICA) is expressed on the cellular surface as a response to stress and is the ligand for natural killer group 2 member D (NKG2D) cells, thus enhancing local cytotoxic immunity. The production of ADAM9 by the neoplastic tissue unties soluble MHC class I-related chain A (MICA), weakening cytotoxic surveillance; this can be restored through sorafenib and regorafenib ADAM9 suppression [45,46]. Elevated ADAM9 expression in HCC patients was associated with a lower survival. The ADAM9-MICA-NKG2D pathway could represent an ideal target for immunotherapy due to the direct correspondence of ADAM9 expression with the presence of inhibitory checkpoint molecules. It has been demonstrated that nivolumab-responding patients showed a significant reduction in ADAM9 messenger-RNA (mRNA) serum levels compared to non-responders. This molecule was undetectable in blood samples of patients with complete response to regorafenib and NK cell immunotherapy [43].

### 3.2. Growth Factors

A crucial field of investigation is the area of circulating biomarkers is undoubtedly represented by pro-angiogenetic cytokines: angiopoietin, hepatocyte growth factor (HGF), leptin, vascular endothelial growth factor (VEGF), platelet-derived growth factor (PDGF), and adhesion molecule-1 [47].

Both angiopoietin-1 (Ang-1) and angiopoietin-2 (Ang-2) interact with the endothelial cell-specific tyrosine kinase receptor (Tie-2). The first enhances vessel stability, while the second increases vascular permeability along with tissue hypoxia and VEGF overexpression leading to neoplastic angiogenesis and the recruitment of proangiogenic myeloid cells [48]. These mechanisms are involved in antiangiogenetic drug resistance and subsequent reduction in patient survival [49].

In the SHARP trial, baseline plasma levels of VEGF and Ang-2 represented independent predictors of survival in HCC patients treated with sorafenib [31,50]. In particular, Ang-2 showed interesting characteristics suggesting its possible application in treatment monitoring since its increase was related with poor outcomes in both the sorafenib and placebo groups [31]. Similarly, in a study performed in 122 patients with advanced HCC treated with sorafenib, Miyahara et al. confirmed that an elevated expression of Ang-2 was associated with a shorter PFS (HR 1.84) and OS (HR 1.95) [51].

Another study compared the plasma levels of Ang-1, Ang-2, and VEGF with the standard AFP in 240 patients with HCC from early to advanced stages. They found that, regardless of HCC etiology, the plasma levels of Ang-2, associated with other prognostic factors such as tumor stage, invasiveness and liver function, were better prognostic marker of OS and PFS compared to AFP, VEGF and Ang-1. The AUROC of baseline Ang-2 levels (0.909) underlined a strong predictive power for 1-year survival compared to the AUROC of Ang-1 (0.535, *p* < 0.001), AFP (0.817, *p* = 0.03) and VEGF (0.577, *p* < 0.001). Similar results were obtained for 2- and 5-years survival. High Ang-2 levels were also an independent factor associated with reduced PFS (HR 1.55, *p* = 0.01) [47].

For years sorafenib has been the only protagonist in advanced HCC treatment, and the majority of molecular biomarkers were studied to match with its therapeutic mechanisms. At present there is still a lack of predictive biomarkers for lenvatinib. Chuma et al. analyzed the levels of several circulating angiogenic factors (VEGF, fibroblast growth factor 19 FGF19, FGF23, and Ang-2) in 74 Child-Pugh-A patients with advanced HCC at baseline and after 4 weeks of treatment with lenvatinib. Responders showed increased FGF19 (ratio vs. baseline: 2.09 in responders and 1.32 in non-responders; *p* = 0.0004) and reduced Ang-2 levels (ratio vs. baseline: 0.584 in responders and 0.810 in non-responders; *p* = 0.0002). No significant differences were detected in FGF23 and VEGF levels. The multivariate analysis confirmed the combination of serum FGF-19 and Ang-2 as an independent predictive factor for therapeutic response (OR 9.143; *p* = 0.0012) and PFS (HR 0.171; *p* = 0.0240) [52].

In a phase 2 study (NCT01246986) with galunisertib (an inhibitor of the transforming growth factor beta, TGF-β) as a second-line treatment for patients with advanced HCC, Giannelli et al. demonstrated an association between a decrease in AFP and TGF-β1 circulating levels and an increase in OS. In particular, AFP responders (patients with a >20% AFP reduction) showed a median OS of 21.5 months versus 6.8 months for non-responders (*p* = 0.0015). Whereas, TGF-β1 responders had a median OS of 11.2 months compared to 5.3 months of non-responders (*p* = 0.0036) [53,54].

Aberrant signaling of the fibroblast growth factor pathway (FGF) and its receptors (FGF-R) has been noticed in several malignancies driving researchers’ attention to these molecular processes as favorable targets for anti-cancer therapies [55]. In particular, the FGF19-FGF4 pathway is amplified in about 5–10% of HCC patients, determining hepatocyte proliferation and sorafenib-resistance [56,57]. This could be applied as a predictor of response to FGFR4 kinase inhibitors that are still under trial investigation, such as H3B-6527 [58], FGF401 [59] and BLU-554 [60].

Finally, the HGF/mesenchymal-epithelial transition factor (c-MET) pathway seems to be involved in HCC development and could predict poor prognosis and sorafenib resistance [61]. For these reasons it was investigated as a possible biomarker.

In a recent study by Kim et al. developed a risk scoring system based on six-covariates (etiology, platelet count, BCLC stage, PIVKA II, HGF and FGF) to identify three groups of risk. Their results confirmed the usefulness of this scoring system in predicting responses to sorafenib and survival with a median OS of 19 months in the low-risk group, 11.2 months in the intermediate, and 6.1 months in the high-risk group (*p* < 0.001) [62].

Moreover, in a subgroup analysis performed in a phase II trial investigating the efficacy of tivantinib (MET-inhibitor) versus placebo as second-line treatment after sorafenib, Santoro et al. showed an association between MET overexpression and poor HCC prognosis [63]. Despite the initial encouraging performance these results were not confirmed by a prospective randomized phase III trial [64,65].

### 3.3. Genetic Biomarkers

Genetic biomarkers include circular RNAs, microRNAs (miRNAs) circulating cell-free DNA (cfDNA). They have been extensively studied for HCC diagnosis, but they have also an important role as prognostic markers in advanced HCC.

#### 3.3.1. Circular-RNA

Circular RNAs are endogenous non-coding RNAs that can be found in human tissues and fluids, representing sponges for mRNA (for transcriptional and translational control) and a regulator of protein expression (modulating their interactions through sequestration or translocation) [66]. Song et al. evaluated the regulatory role of circular RNAs in oncogenic processes, particularly, analyzing the expression of hsa_circ_0001821 in the plasma of patients with colorectal cancer, lung cancer and HCC. They concluded that this biomarker could be used as pan-cancer marker with high diagnostic value. In particular, in patients with HCC, this marker seemed to be significantly related to tumor size and advanced stages [67].

#### 3.3.2. Micro-RNA

MicroRNAs are endogenous, non-coding molecules of RNA with regulatory roles that, through their interactions with mRNAs, are involved in cell proliferation and apoptosis, and consequently in carcinogenesis, invasion, tumor progression and metastasis in different kinds of tumors [68]. They can be found in neoplastic tissue, but also in plasma and serum in an extraordinarily stable form (preserved from the degradation by endogenous RNase) and for this reason, they attract the attention of researchers as a potential diagnostic/prognostic biomarker [69].

Li et al. analyzed the serum levels of four circulating miRNAs (miR), hyperexpressed in HCC. Among them, miR-221 was associated with cirrhosis, tumor size, HCC stage (*p* = 0.003, *p* < 0.001, *p* = 0.016, respectively) and HCC prognosis. In particular OS was significantly reduced in patients with high miR-221 levels (27.6% vs. 62.3%, *p* < 0.05). The multivariate Cox regression analysis demonstrated that cirrhosis, tumoral size and high serum levels of miR-221 were independent predictors of OS [70]. In another study lower pre-treatment levels of miR-221 were associated with a better response to sorafenib (*p* = 0.007) [71].

Zhou et al. found a significant association between studied miR-93e5p and the prognosis of HBV-related HCC. Its plasmatic and urinary dosage were useful for early detection of HCC, had an important role in predicting PFS after curative hepatectomy in early HCC, and OS in advanced HCC undergoing non-curative treatments [72].

Another study highlighted the potentially prominent role of miR-219-5p as a prognostic biomarker in advanced HCC. This miR is a crucial post-transcriptional downregulator of NEK6 (never in mitosis gene a-related kinase 6) gene expression with a subsequent dysregulation in β-catenin/c-Myc-associated gene expression. MiR-219-5p seems to decrease in HCC with a subsequent overexpression of NEK6, and an enhanced consequent cancer progression demonstrated in both in vitro and in vivo cellular proliferation models. Moreover, elevated NEK6 expression has been related to a worse outcome and prognosis in stage III-IV HCC [73].

Finally, the most relevant tissue miRs evaluated in patients with advanced HCC treated with sorafenib were miR-224 and miR-425-3p. The first has been related to longer OS (HR = 0.24, *p* = 0.012) and PFS (HR = 0.28, *p* = 0.029) [74], and the second to better PFS (HR=0.5, *p* = 0.007) and time to progression (TTP) (HR = 0.4, *p* = 0.0008) [75].

#### 3.3.3. Cell-Free DNA

Fragmented cfDNA can be found in blood samples after cell apoptosis and their levels are higher in patients with cancer [76]. For this reason, they can be useful in identifying specific neoplasia-related genetic mutations related to disease progression. In particular, mutations in the human telomerase reverse transcriptase (hTERT) gene are the most frequently identified in patients with HCC and could become an important resource in disease progression monitoring [77]. Telomerase is an essential enzyme contributing to DNA stability that can be activated by epigenetic mechanisms or through somatic mutations in the TERT promoter that occurs in many solid tumors. In a study performed by Muraoka et al. on 67 patients with HCC undergoing TACE and chemotherapy with tyrosine-kinase inhibitors (TKI), blood levels of hTERT promoter mutant DNA were useful in revealing HCC volume and to predict long-term therapy responsiveness (Table 1) [78].

In the case of TACE, blood hTERT levels were related to the amount of tumor necrosis (*p* < 0.001), whereas in patients treated with TKI exhibited a peak of cfDNA levels at 1 week after the beginning of therapy and were able to predict PFS and long-term responses to therapy. These results were confirmed by a small retrospective study by Hirai et al. on 133 patients with advanced stage HCC undergoing TKI or TACE. The results reported 54.6% of patients presented with mutations in the TERT promoter that were associated with large tumor size, high serum levels of DCP, and a shorter OS [79]. This mutation should be further investigated to understand its prognostic value because its detection is undoubtedly associated with rapid cellular proliferation and subsequent poor prognosis, thus representing an interest target for liquid biopsy.

It is clear that circulating tumor DNA may offer an extremely precise update on HCC poor differentiation.

Dong et al. quantitatively analyzed circulating cfDNA and tumor fraction in plasma samples from 266 patients with a prognostic purpose. Modifications in the tumor fraction during TACE were related to neoplastic burden and represented a prognostic predictor of progression and a reliable index of response to treatment, thus representing an alternative to commonly used biomarkers such as AFP and the mRECIST system [80]. The difference between pre- and post-TACE tumor fraction levels, measured during the first TACE session, was the ability to predict PFS, even in patients undergoing repeated treatments. Post-TACE tumor fraction assessed the remaining tumor burden and was detectable in 81.3% of HCC patients compared to AFP that was detected in 70.3%. A higher post-TACE tumor fraction was related to a reduced OS (149 vs. 660 days; *p* < 0.001) and PFS (69 vs. 192 days; *p* < 0.001). The analysis of genomic copy number amplification and loss showed that chromosome 16q amplification was associated with reduced TACE response. Similarly, the amplification of copy number of chromosomes 1q, 3p, 6p, 8q, 10p, 12q, 18p or 18q decreased lipiodol deposition and subsequent TACE efficacy. The authors concluded that, cfDNA low-depth sequencing could represent a non-invasive and relatively low-cost method to acquire important prognostic information about neoplastic burden and quantify TACE response in order to guide clinical decisions and allow a personalized therapeutic approach. They remarked on the importance of post-TACE tumor fraction monitoring as a non-invasive detector for neoplastic relapse before CT/MRI detectable signs appeared. However, large-scale studies are certainly needed in order to validate these interesting and promising findings.

### 3.4. Neutrophil-to-Lymphocyte Ratio

Neutrophil-to-lymphocyte ratio (NLR) is a simple biochemical parameter initially proposed as a marker of inflammatory status and disease-related inflammation (Figure 1) [81]. It has been used as a prognostic marker in several cancers [82]. Regarding HCC, the NLR has been extensively studied for its value in predicting mortality and poor therapeutic response in each stage of the disease [83,84,85]. This feature could be related to an alteration of the inflammatory microenvironment of the tumor, which can result in tumor growth or recurrence [86]. Furthermore, the increase in NLR has been associated with tumor expression of cytokeratin 19 (CK19) [87], and consequently a more aggressive tumor behavior with weaker prognosis [88,89].

The NLR showed a positive correlation with HCC recurrence after liver transplantation [90], and its pre-treatment values are predictive of poor prognosis in patients with intermediate HCC undergoing TACE [91].

In a retrospective analysis performed in patients with advanced HCC (BCLC stage C) treated with glass microsphere radioembolization, the post-treatment increase of NLR was associated with a reduced OS, although pre-treatment values did not show a significant association [92].

In patients with advanced HCC, the NLR predicted not only a poor prognosis, but also a worse response to systemic therapy. In a retrospective study by Bruix et al. higher pre-treatment NLR values, together with high AFP and macroscopic vascular invasion, were associated with a worse response to sorafenib and reduced OS [93]. Similar results were obtained by Eso et al. in 40 patients treated with atezolizumab-bevacizumab reporting a reduced PFS was observed with NLR > 3.21 (*p* < 0.0001) [94].

The study by Muhammed et al. focused on 362 patients undergoing ICI for advanced HCC, and the univariate and multivariate analysis identified NLR as an independent prognostic factor for OS (HR 1.95, *p* < 0.001; HR 1.73, *p* = 0.002, respectively). In particular, a NLR ≥ 5 was associated with a lower OS (7.7 vs. 17.6 months; *p* < 0.0001), PFS (2.1 vs. 3.8 months, *p* = 0.025), and ORR (12 vs. 22%; *p* = 0.034) [95].

These results were confirmed by Wang et al. in a retrospective study on 171 HCC patients treated with apatinib, reporting lower pre-treatment levels of NLR were associated with longer OS and PFS [96].

Finally, in a recent single prospective controlled study, the NLR showed an independent prognostic value within each disease stage and, in particular, predicted HCC mortality in the terminal phase of the disease [97].

Based on these findings, the NLR could represent a simple parameter to select patients with advanced HCC with higher probability of a poor response to systemic therapy and consequently poor prognosis. However, more detailed studies are necessary to find the biological processes that justify this outcome, especially in the field of ICI.

## 4. Tissue Biomarkers

### 4.1. Proteoglycans

Proteoglycans, a group of highly glycosylated proteins mainly located in cell membranes and the extracellular matrix, are implicated in the modulation of neoplastic progression due to their role in signaling activities that can influence the availability of growth factors involved in stromal environment remodeling, angiogenesis and tissue regeneration (Figure 1) [98]. Among them, serglycin has been associated with advanced-stage HCC, vascular invasion and bone-metastases, and reported as an independent predictor of OS and time to recurrence (TTR) [18]. The same molecule has been included in a diagnostic model comprising seven peptides with an elevated predictive power and detection rate for the presence of HCC bone metastatic lesions [99].

Another proteoglycan, glypican-3 (GPC3), involved in the Wnt and Hedgehog signaling pathways has been related to a worse prognosis in patients with HCC [100,101]. On the other hand, the recently developed anti-GCP3 antibody (GC33) demonstrated an important anti-tumor effect in patients with HCC, enhancing cytotoxic T lymphocytes (CTLSs) infiltration and activity in neoplastic tissues [102,103]. Several studies suggested the role of syndecan-1, a cell surface proteoglycan, as a potential prognostic biomarker of advanced HCC. Its serum hyperexpression and tissue-dosage reduction seem to be related to scarcely differentiated HCC and elevated extrahepatic metastatic potential [104,105]. These characteristics are linked to the complex that syndecan-1 forms with insulin-like growth factor (IGF-1) receptor and ανβ integrin, that has a major role in tumorigenesis and angiogenesis. In a recent pre-clinical trial, synstatin, an inhibitor of this complex, showed promising results with the reduction of angiogenic growth factors (basic-FGF and VEGF) in a rat model of HCC [106].

### 4.2. Stem Cells

Kim et al. analyzed cancer stem cells genes in histological samples derived from tumor biopsies (EpCAM, CK8, SALL4, ALDH1A1, CD13, CD24, CD44, CD90, CD133, ALB, AFP) as prognostic biomarkers in 47 patients with advanced HCC treated with sorafenib. They found that overexpression of CD133 and CD90 was linked to a poor therapeutic response. In particular the combination of CD133 and CD90 overexpression was associated with a reduced PFS (2.7 vs. 5.5 months, *p* = 0.04) suggesting a possible role of these genes as future biomarkers in the prediction of HCC response to chemotherapy (Figure 1) [107].

### 4.3. Organoid Cultures

The organoids (tumoroids/organoid cultures) derived from primary liver cancers have an enormous potential as prognostic biomarkers in drug screening or resistance detection, due to their characteristic mirroring of parental tumor histology, genetic and transcriptome patterns (Figure 1) [108]. However, the absence of stromal elements and the immune system limit the possibilities to investigate the correlation between neoplastic cells and the surrounding stroma or immune system. Further studies are necessary to definitely confirm these brilliant in vitro results and allow their application as prognostic biomarkers for a personalized treatment approach in everyday practice.

## 5. Imaging Biomarkers

An accurate evaluation of tumor response after therapy has become an essential part in the management of advanced HCC not only for determining treatment efficacy, but also for subsequent therapeutic planning and as a marker for survival. Traditionally, the European Association for the Study of the Liver (EASL) guidelines recommend that the evaluation criteria for tumor progression or remission should incorporate measurements of viable enhancing areas of the tumor [109]. The modified response evaluation criteria in solid tumors (mRECIST) are currently the primary criteria for evaluating therapeutic efficacy in solid tumors [110].

In a phase II study of brivanib in advanced HCC, mRECIST was able to demonstrate a higher response and disease control rate and a longer time to progression than the classical volume-based criteria [111]. Similarly, in a retrospective study of HCC treated with sorafenib, patients categorized as responders according to mRECIST had a longer OS than non-responders [112].

However, mRECIST criteria rarely reflect the effect of therapies that result in decreased enhancement without necrosis, typical of molecular targeted agents. In this clinical setting, it would be beneficial to measure tumor perfusion, because vascular changes can happen long before there is any evidence of tumor volume variation on conventional imaging. As a result, perfusion imaging techniques such as dynamic contrast-enhanced (DCE) ultrasound (US), CT or MRI begin to play a critical role in the evaluation of antiangiogenetic therapies [113]. All these techniques enable quantification of tumor vascularity by measuring the temporal changes in tissue enhancement following intravenous contrast administration. A variety of imaging protocols have been proposed for perfusion imaging, and the computed perfusion parameters are dependent on the scan protocol and the software for image processing [113].

On perfusion CT, HCC has been reported to show substantially higher perfusion (high blood flow, [BF], blood volume [BV] and permeability surface area [PS] with low mean transit time [MTT]) compared to normal liver tissue [114]. Jiang et al. demonstrated that HCC with a higher baseline MTT correlated with a favorable clinical outcome [115].

After antiangiogenic drugs or HCC directed therapies, a decrease in BF and BV has been shown within days of initial treatment [114,115]. In a recent paper by Ippolito et al., CT perfusion parameters obtained before and every 2 months after sorafenib administration were compared between non-progressor (complete response, stable disease or partial response) and progressor (progressive disease) groups according to mRECIST [116]. A higher survival rate was observed in the non-progressor group compared to the progressor (48.6% vs. 28.6%), and a statistically significant correlation (*p* = 0.01) was found between percentage variation of perfusion parameters, from baseline to follow-up, and overall survival rate [116].

Similarly, Liang et al. reported that signal parameters of DCE-MRI over tumor and liver parenchyma correlated with tumor response and survival in HCC patients receiving a combination of radiotherapy with an anti-angiogenic agent [117]. In a recent paper an early reduction in tumor perfusion detected by DCE-MRI biomarkers may predict survival outcomes in patients with advanced HCC under second-line targeted therapy following sorafenib failure [118].

Among different contrast enhanced imaging techniques, dynamic contrast-enhanced ultrasonography (DCE-US) has emerged as a versatile tool for monitoring anti-angiogenetic treatments since it is a non-invasive method that easily allows repeated examinations and provides both morphological and functional data. Several studies demonstrated that DCE-US can be used to predict tumor responses and patient survival in HCC patients treated with antiangiogenetic agents [119,120]. In particular, we demonstrated that the percentage variation in three DCE-US parameters (area under the curve [AUC], peak enhancement [PI], and slope of wash-in [Pw]) after 2 weeks of treatment with sorafenib significantly correlated with response (*p* = 0.002, <0.001 and 0.003 respectively). Moreover, a decrease of AUC and an increased/unchanged value of time to peak (TP) and MTT were associated with longer survival (*p* = 0.045, 0.029 and 0.010, respectively) [120].

A variety of imaging protocols, other than perfusion techniques have been proposed as biomarkers in patients with advanced HCC.

A recent study demonstrated that baseline imaging features showing aggressive tumor biology and in particular, satellite lesions, atypical HCC, peritumoral arterial enhancement, and larger lesion size, can serve as imaging biomarkers for OS and liver decompensation in patients receiving both sorafenib and selective internal radiation therapy for HCC [121].

Tumor stiffness and fibrosis could be another factor in predicting the therapeutic response in patients with advanced HCC since the abnormal cellular microenvironment of neoplastic conditions can be associated with increased stiffness [122].

Magnetic resonance elastography (MRE) is a relatively novel technique that has been shown to be superior to US-based elastography for the assessment of liver stiffness (LS) [123]. In a study performed by Kim et al., higher MRE-assessed LS was a potential biomarker for predicting poor OS (HR 1.54, *p* < 0.001) and significant liver injury in advanced HCC patients treated with sorafenib (HR 1.62, *p* = 0.001) [124].

Similar results were obtained by Qayyum et al. in 9 HCC patients treated with pembrolizumab. Early changes in tumor stiffness evaluated by MRE were correlated significantly with OS (R = 0.81) and time to progression (R = 0.88, *p* < 0.01). Moreover, stiffness was significantly correlated with intratumoral T lymphocytes in tumor biopsies [125].

Liver Imaging Reporting and Data System (LI-RADS) is a standardized diagnostic system for the analysis of CT and MR imaging features in patients at risk of HCC development [126]. Recent data from small-scale studies highlighted the potential prognostic role of LI-RADS classification in the detection of HCC biologic aggressiveness (microvascular invasion, histological characteristics) that can directly influence the clinical outcome [127]. Fat in mass and hyperintensity in the hepatobiliary phase represents favorable prognostic biomarkers in the LI-RADS score, associated with a low risk of HCC recurrence after treatment [127,128,129]. Signs of poor prognosis with a higher risk of HCC recurrence, disease progression and shorter survival, are represented by rim arterial-phase hyperenhancement, corona enhancement, non-smooth tumor margins, LR-M HCC, low apparent diffusion coefficient, and peritumoral hypointensity in the hepatobiliary phase [130,131,132,133]. A concrete demonstration of the importance of the LI-RADS classification with prognostic intent on postoperative outcomes was reported by Centonze et al. in a study involving 186 HCC patients (53 LR-3/4,133 LR-5). Here, remarkable percentages of satellitosis (9.4% vs. 25.8%), capsular infiltration (11.3% vs. 28%) and microvascular invasion (22.6% vs. 41.7%) were described both in LR-3/4 and LR-5 groups, respectively. For this reason, a cautious radiological assessment is fundamental before treatment planning in intermediate-risk subjects [134].

A precise association between imaging biomarkers and prognostic factors is still difficult, but useful for a rudimentary definition of predictive imaging features. It could be useful to include these characteristics into an algorithm alongside morphologic criteria. Nevertheless, further validation studies are required in order to increase its potential effectiveness in HCC management, patients’ outcome improvement, and shaping individualized treatment plans and follow-up.

It is known that tumor heterogeneity is closely related to tumor prognosis, most notably in HCC lesions. The recent developed radiomics technique can obtain intratumoral heterogeneity in a non-invasive way that is relevant to patient prognosis. This is an emerging field in which high dimensional features are mathematically extracted from radiological images with subsequent conversion of images into mineable data [135]. Recently, radiomic analyses of HCC using CT and MRI images has been shown to have a high prediction accuracy [136,137].

A multi-feature-based radiomic signature was identified to be an independent biomarker for OS and TTP in patients with advanced HCC treated with apatinib plus TACE. The combined use of a radiomic signature and AFP in the clinical-radiomic nomograms performed better than radiomic nomograms alone [138].

In a recent retrospective study, a radiomic-based model for predicting β-arrestin1 phosphorylation in HCC was developed using visual imaging features on preoperative CT images [139]. Imaging and radiomic features were combined to establish clinico-radiological (CR) and clinico-radiological-radiomic (CRR) models by using multivariable logistic regression analysis. The authors found that the CRR model integrating the radiomics score with clinico-radiological risk factors showed a better discriminative performance (AUC = 0.898, 95%CI, 0.820 to 0.977) than the CR model (AUC = 0.898 vs. 0.794, *p* = 0.011), for predicting β-arrestin1 phosphorylation-positive HCC and OS [139].

Although promising, none of these functional imaging biomarkers have undergone enough standardization and validation to be used in clinical practice. Further studies are warranted to determine unclear aspects such as optimal timing and best quantitative dynamic parameter to assess responses to HCC treatment.

## 6. Conclusions

A large number of prognostic biomarkers for advanced HCC have been reported; however, their clinical applications are still far away in comparison to other cancers (Table 2). Serum biomarkers could improve HCC management in two ways: first, they can provide a deeper knowledge of the tumor genetic landscape; second, they can be useful for tumor surveillance allowing serial sampling over time. In particular, the dynamic evaluation of all the mentioned biomarkers could have a critical role for choosing the appropriate therapy and for evaluating treatment response, tumor progression, drug resistance and cancer recurrence.

Due to the high heterogeneity of HCC, further research is necessary to validate the potential of non-invasive biomarkers in the management of advanced tumors.

## Figures and Tables

**Figure 1 cancers-14-04647-f001:**
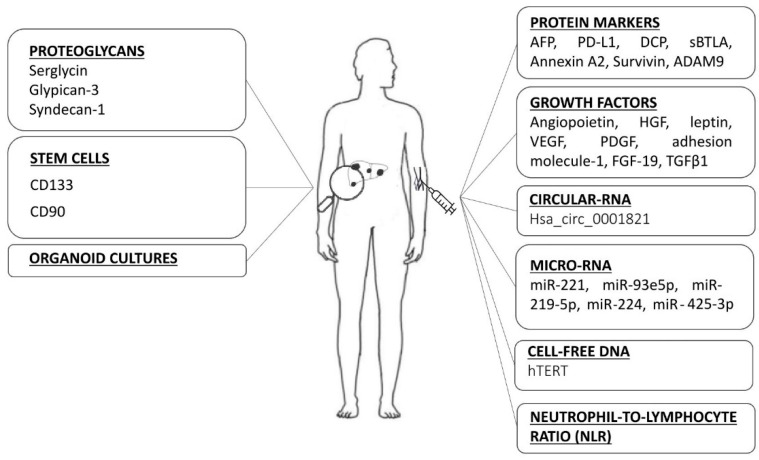
Many heterogeneous categories of biomarkers are being investigated as possible prognostic parameters in predicting the therapeutic efficacy and survival in advanced HCC. AFP: alpha-fetoprotein, PD-L1: programmed cell death-1 ligand, DCP: des-γ-carboxy prothrombin, sBTLA: soluble B and T lymphocyte attenuator, ADAM9: a-disintegrin-and-a-metalloprotease-9, HGF: hepatocyte growth factor, VEGF: vascular endothelial growth factor, PDGF: platelet-derived growth factor, FGF-19: fibroblast growth factor 19, TGF-β1: transforming growth factor beta 1, miR: microRNA, hTERT: human telomerase reverse transcriptase.

**Table 1 cancers-14-04647-t001:** The prognostic role of clinical biomarkers in patients with advanced hepatocellular carcinoma.

Article	Patients (Number)	Therapy	Biomarker	Prognostic Data
Giannelli G [53]	149	Galunisertib: phase 2 study (NCT01246986)	AFP TGF-β1	OS: Group A: 7.3 months (95% CI: 4.9–10.5) Group B: 16.8 months (95% CI: 10.5–24.4)
		Group A: baseline AFP > 1.5 ULN Group B: baseline AFP < 1.5 ULN		AFP responders (21% patients in group A; >20% AFP reduction): median OS 21.5 months; AFP non-responders: 6.8 months (*p* = 0.0015). TGF-β1 responders (51% of all patients): median OS 11.2 months; AFP non-responders 5.3 months (*p* = 0.0036).
Gyöngyösi B [74]	20	Sorafenib	Tissue miR-224	OS (HR = 0.0.24, 95%CI: 0.07–0.79, *p* = 0.012) PFS (HR = 0.28, 95%CI: 0.09–0.92, *p* = 0.029)
Kelley RK [29]	707	Cabozantinib vs. placebo	AFP	Median OS cabozantinib versus placebo: Baseline AFP < 400 ng/mL: 13.9 versus 10.3 months [HR, 0.81; 95% confidence interval (CI), 0.62–1.04] Baseline AFP ≥ 400 ng/mL: 8.5 versus 5.2 months (HR, 0.71; 95% CI, 0.54–0.94)
				Week 8 AFP response rate: 50% vs. 13% (cabozantinib vs. placebo)
				Median OS (cabozantinib arm): 16.1 versus 9.1 months (HR, 0.61; 95% CI, 0.45–0.84) with and without AFP response.
Kim HY [62]	124	Sorafenib	PIVKA II, HGF, FGF	OS (*p* < 0.001): 19.0 months (low-risk group); 11.2 months (intermediate); 6.1 months (high-risk group)
Lee PC [35]	95	Nivolumab or pembrolizumab	AFP	AFP reduction >10%: ORR 63.6% vs. 10.2% (*p* < 0.001); DCR 81.8% vs. 14.3% (*p* < 0.001) >20%: ORR 64.7% vs. 14.8% (*p* < 0.001); DCR 82.4% vs. 20.4% (*p* < 0.001) >30%: ORR 61.5% vs. 19.0% (*p* = 0.001); DCR 84.6% vs. 24.1% (*p* < 0.001)
Li J [70]	46	NA	miR-221	OS: 27.6% versus 62.3% (high miR-221 versus low miR-221 expression; *p* < 0.05)
Llovet JM [31]	602	Sorafenib vs. placebo	VEGF-A Ang-2	Median survival (low versus high baseline VEGF-A): 10 versus 6.2 months Median survival: 14.1 and 6.3 months (low versus high baseline Ang2)
Miyahara K [51]	122	Sorafenib	Ang-2	PFS (Ang-2: HR 1.84; 95%CI 1.21–2.81) OS (Ang-2: HR 1.95; 95%CI 1.21–3.17)
Muraoka M [78]	67	TACE (32 patients) Sorafenib (6 patients) Lenvatinib (29 patients)	Cell-Free Human hTERT mutant DNA	Median survival times: Positive for mutant DNA → 11.9 months Negative for mutant DNA → 20.2 months
Shao YY [30]	72	Sorafenib or bevacizumab or thalidomide in combination with metronomic 5-fluoropyrimidine	AFP (Responders vs. non-responders)	ORR 33% vs. 8% (*p* = 0.037) DCR: 83% vs. 35% (*p* = 0.002) PFS: 7.5 vs. 1.9 months (*p* = 0.001) OS: 15.3 vs. 4.1 months (*p* = 0.019)
Vaira V [75]	26	Sorafenib	miR-425-3p	PFS (HR = 0.5, 95%CI: 0.3–0.9, *p* = 0.007) TTP (HR = 0.4, 95%CI: 0.2–0.7, *p* = 0.0008)
Zhu AX [33]	292	Ramucirumab versus placebo	AFP (≥400 ng/mL)	OS (8.5 vs. 7.3 months; HR 0.71, 95% CI 0.53, 0.95; *p* = 0.0199) PFS (2.8 vs. 1.6 months; HR 0.452, 95% CI 0.34, 0.60; *p* < 0.0001)

AFP: alpha-fetoprotein, Ang2: angiopoietin-2, CI: confidence interval, DCR: disease control rate, FGF: fibroblast growth factor, HGF: hepatocyte growth factor, HR: hazard ratio, miR: microRNA, ORR: objective response rate, OS: overall survival, PFS: progression free survival, PIVKA II: protein induced by vitamin K absence-II, TACE: transarterial chemoembolization, hTERT: human telomerase reverse transcriptase, TGF-β1: transforming growth factor beta1, TTP: time to progression, ULN: upper limit of normal, VEGF-A: vascular endothelial growth factor-A.

**Table 2 cancers-14-04647-t002:** The advantages and disadvantages of the principal biomarkers applied in the management of hepatocellular carcinoma.

	Advantages	Disadvantages
**Serum Biomarkers Non-invasive**		
*1. Protein biomarkers*	*Alpha-fetoprotein:* main biomarker for diagnosis, prognosis and evaluation of HCC therapeutic response [9] prognostic value also for new systemic therapies [33,34,35]	*Alpha-fetoprotein:* limited sensitivity and specificity; false positives or false negatives according to the cut-off or HCC stage [23]; elevated in other conditions [23] Further studies are needed
	*Other protein biomarkers (PD-L1, DCP, sBTLA, annexin A2, ADAM9):* *PD-L1:* may predict response to immune checkpoint inhibitors [35] *DCP:* may have a better prognostic value than AFP in detecting large tumors, poor differentiated HCC or PVT [37,38] Possible predictors of HCC at advanced stage [39,40] or response to immunotherapy [43]	*DCP:* seems less effective for small HCCs [37]
*2. Growth factors*	*Ang-1, Ang-2, VEGF, FGF-19, MET:* Potential predictors of OS and response to anti-angiogenic therapy [47,48,49,50,51,52,53,54,55,56,57,58,59,60,61,62,63,64,65]	Further studies are needed
*3. Genetic biomarkers*	*Circular RNAs, microRNAs circulating cell-free DNA:* Potential predictors of advanced HCC, overall OS and response to systemic therapy [67,68,69,70,71,72,73,74,75,76,77,78,79,80]	Further studies are needed
*4. Neutrophil-to-lymphocyte ratio*	Simple, cheap and obtainable from routinary analysis [81] May predict overall OS in different stages of the disease [83,84,85,97]	Further studies are needed Less evidence in patients undergoing systemic therapy [93,94]
**Tissue Biomarkers**		
*1. Proteoglycans*	*Serglycin, GPC3, syndecan-1:* Potential predictors of advanced HCC and overall OS [18,100,101,102,103,104,105,106] Potential predictors of bone metastasis [18]	Data mainly from murine models Further studies are needed
*2. Cancer stem cells*	Potentially useful to identify molecular biomarkers of response to therapy or prognosis [107]	Further studies are needed
*3. Organoid Cultures*	Potentially useful to test drug sensibility or to identify other biomarkers [108]	Impossibility to reproduce tumoral stroma Further studies are needed
**Radiological Biomarkers**		
*mRECIST criteria*	Primary criteria for evaluating therapeutic efficacy in solid tumors [110] Evaluation of response to treatments in advanced HCC: better than classical volume-based criteria [111,112].	Poor definition of vascular changes and therapeutic effects in course of anti-angiogenic therapy
*Perfusion imaging techniques: dynamic contrast-enhanced (DCE) CT or MRI*	Critical role in the evaluation of response to antiangiogenetic therapies [113] May independently predict clinical outcomes [114,115] May predict clinical response to systemic therapy [114,115,116,117,118]	High costs Not always available Risk of contrast-induced nephropathy
*Perfusion imaging techniques: dynamic contrast-enhanced (DCE) ultrasound (US)*	Cheaper than CT or MRI Easily repeatable No risk of contrast-induced nephropathy	Not always available Less evidence than CT or MRI Cannot examine total liver parenchyma
*Imaging features of tumor biological aggressiveness at diagnosis (satellite lesions, atypical HCC, peritumoral arterial enhancement, larger lesion size)*	Prediction of response to systemic therapy or selective internal radiation therapy [121]	Poor data regarding therapeutic response to anti-angiogenic or immune therapies Further studies are needed
*Tumor stiffness measured with MRI-Elastography*	Prediction of OS and therapeutic response to systemic therapy [123,124,125]	High cost Not always available Further studies are needed
*Radiomic signatures*	Potentially a new, independent biomarker of prognosis, OS and therapeutic response [125,126,127,128,129,130]	Recent technique, still not available outside of highly-specialized centres Further studies are needed
*LI-RADS*	Detection of HCC biologic aggressiveness (microvascular invasion, histological characteristics) directly influencing clinical outcome [127]	Still lacking a precise association between imaging biomarkers and prognostic factors Further validation studies required

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
