# Peer review of "Prognostic Role of Molecular and Imaging Biomarkers for Predicting Advanced Hepatocellular Carcinoma Treatment Efficacy"

_cancers, 2022, doi:10.3390/cancers14194647_

Round 1
Reviewer 1 Report
This is a well written, comprehensive review focusing on serum and radiological biomarkers for HCC staging and prognostic evaluation
Considering its increasing incidence and high mortality, the management of HCC represent a major challeng in clinical oncology
The prognostic models delevoped so far that incorporates AFP and tumor burden accordin to oreoperative imaging (such as Metroticket in LT setting) have reached a good performance, but there is still room for improvement, and implementation in liquid biopsy will probably play a crucial role in next future
Focusing on radiology, the development of AI and radiomics and their application in clinical practice will represent another pivotal pillar for increase the ability to predict the prognosis in HCC
Waiting for introduction of radiomics in clinical practice, there are recent evidences that suggest a possible role of LIRADS as a prognostic indicator, that should be added in the paper (10.1111/liv.15362; 10.3390/diagnostics12010160 )
Best regards
Author Response
We thank the reviewes for these suggestions. We commented the attached manuscript lines 568-585.

Reviewer 2 Report
In this review Cerrito and collegues revise the knowledge on the use of biomarkers with a predictive and prognostic role in advanced HCC. The review is well written and clearly describes the different biomarkers under study in HCC.
Minor revision:
- You describe the use of alpha-fetoprotein (AFP) as the only recognized tool for HCC with predictive and prognostic roles. However, you have not clearly described why its role is debated, and why there is the real need to improve and develop new biomarkers in HCC. For example, the limits of the use of AFP should be explained better (for example in line 105). Then, you state that “several attempts were made in the past to find other potential biomarkers to define response to treatment in advanced HCC […].” (line 183), but you don’t explain the reason underlying the need of such attempts. This aspect should be better described and discussed.
- It could be useful to summarize in a table the advantages and disadvantages of each biomarker described in the text.
Author Response
We would like to thank the Reviewer for the comments and the suggestions. The comment about the limitations of AFP was added at lines 105-112. The reasons underlying the attempts to study new biomarkers was developed at lines 192-197. The table summarizing advantages and disadvantages of biomarkers can be found at pages 16-18.